# AND THE BIT GOES DOWN: REVISITING THE QUANTIZATION OF NEURAL NETWORKS

**Pierre Stock[1,2], Armand Joulin[1], Rémi Gribonval[2], Benjamin Graham[1], Hervé Jégou[1]**
[1]Facebook AI Research, [2]Univ Rennes, Inria, CNRS, IRISA

## ABSTRACT

In this paper, we address the problem of reducing the memory footprint of convolutional network architectures. We introduce a vector quantization method that aims at preserving the quality of the reconstruction of the network outputs rather than its weights. The principle of our approach is that it minimizes the loss reconstruction error for *in-domain* inputs. Our method only requires a set of unlabelled data at quantization time and allows for efficient inference on CPU by using byte-aligned codebooks to store the compressed weights. We validate our approach by quantizing a high performing ResNet-50 model to a memory size of 5 MB ($20\times$ compression factor) while preserving a top-1 accuracy of $76.1\%$ on ImageNet object classification and by compressing a Mask R-CNN with a $26\times$ factor.[1]

## 1    INTRODUCTION

There is a growing need for compressing the best convolutional networks (or ConvNets) to support embedded devices for applications like robotics and virtual/augmented reality. Indeed, the performance of ConvNets on image classification has steadily improved since the introduction of AlexNet (Krizhevsky et al., 2012). This progress has been fueled by deeper and richer architectures such as the ResNets (He et al., 2015) and their variants ResNeXts (Xie et al., 2017) or DenseNets (Huang et al., 2017). Those models particularly benefit from the recent progress made with weak supervision (Mahajan et al., 2018; Yalniz et al., 2019; Berthelot et al., 2019). Compression of ConvNets has been an active research topic in the recent years, leading to networks with a $71\%$ top-1 accuracy on ImageNet object classification that fit in 1 MB (Wang et al., 2018b).

In this work, we propose a compression method particularly adapted to ResNet-like architectures. Our approach takes advantage of the high correlation in the convolutions by the use of a structured quantization algorithm, Product Quantization (PQ) (Jégou et al., 2011). More precisely, we exploit the spatial redundancy of information inherent to standard convolution filters (Denton et al., 2014). Besides reducing the memory footprint, we also produce compressed networks allowing efficient inference on CPU by using byte-aligned indexes, as opposed to entropy decoders (Han et al., 2016).

Our approach departs from traditional scalar quantizers (Han et al., 2016) and vector quantizers (Gong et al., 2014; Carreira-Perpiñán & Idelbayev, 2017) by focusing on the accuracy of the activations rather than the weights. This is achieved by leveraging a weighted $k$-means technique. To our knowledge this strategy (see Section 3) is novel in this context. The closest work we are aware of is the one by Choi et al. (2016), but the authors use a different objective (their weighted term is derived from second-order information) along with a different quantization technique (scalar quantization). Our method targets a better *in-domain* reconstruction, as depicted by Figure 1.

Finally, we compress the network *sequentially* to account for the dependency of our method to the activations at each layer. To prevent the accumulation of errors across layers, we guide this compression with the activations of the uncompressed network on unlabelled data: training by distillation (Hinton et al., 2014) allows for both an efficient layer-by-layer compression procedure and a global fine-tuning of the codewords. Thus, we only need a set of unlabelled images to adjust the codewords. As opposed to recent works by Mishra & Marr (2017) or Lopes et al. (2017), our distillation scheme is sequential and the underlying compression method is different (PQ vs. scalar).

---

[1]Code and compressed models: `https://github.com/facebookresearch/kill-the-bits`.

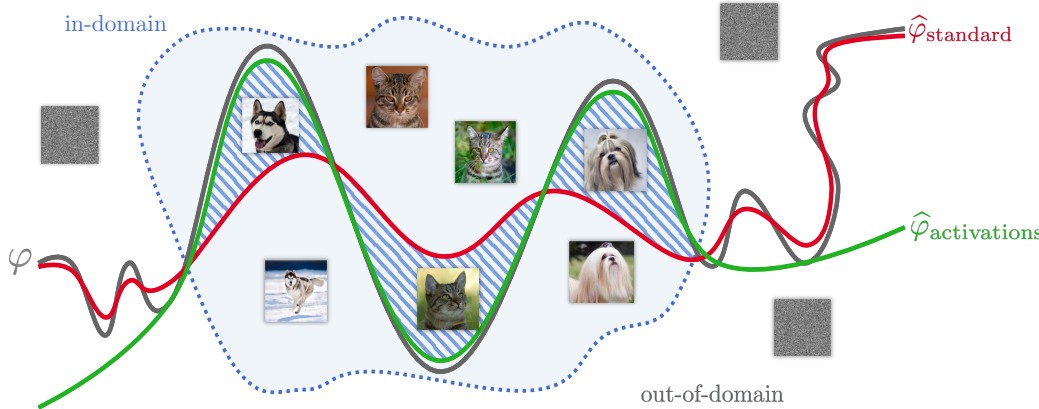

Figure 1: Illustration of our method. We approximate a binary classifier $\varphi$ that labels images as dogs or cats by quantizing its weights. **Standard method**: quantizing $\varphi$ with the standard objective function (1) promotes a classifier $\widehat{\varphi}_{\text{standard}}$ that tries to approximate $\varphi$ over the entire input space and can thus perform badly for in-domain inputs. **Our method**: quantizing $\varphi$ with our objective function (2) promotes a classifier $\widehat{\varphi}_{\text{activations}}$ that performs well for in-domain inputs. Images lying in the hatched area of the input space are correctly classified by $\varphi_{\text{activations}}$ but incorrectly by $\varphi_{\text{standard}}$.

We show that applying our approach to the semi-supervised ResNet-50 of Yalniz *et al.* (Yalniz et al., 2019) leads to a 5 MB memory footprint and a 76.1% top-1 accuracy on ImageNet object classification (hence 20× compression vs. the original model). Moreover, our approach generalizes to other tasks such as image detection. As shown in Section 4.3, we compress a Mask R-CNN (He et al., 2017) with a size budget around 6 MB (26× compression factor) while maintaining a competitive performance.

## 2 RELATED WORK

There is a large body of literature on network compression. We review the works closest to ours and refer the reader to two recent surveys (Guo, 2018; Cheng et al., 2017) for a comprehensive overview.

**Low-precision training.** Since early works like those of Courbariaux et al. (2015), researchers have developed various approaches to train networks with low precision weights. Those approaches include training with binary or ternary weights (Shayer et al., 2017; Zhu et al., 2016; Li & Liu, 2016; Rastegari et al., 2016; McDonnell, 2018), learning a combination of binary bases (Lin et al., 2017) and quantizing the activations (Zhou et al., 2016; 2017; Mishra et al., 2017). Some of these methods assume the possibility to employ specialized hardware that speed up inference and improve power efficiency by replacing most arithmetic operations with bit-wise operations. However, the back-propagation has to be adapted to the case where the weights are discrete.

**Quantization.** Vector Quantization (VQ) and Product Quantization (PQ) have been extensively studied in the context of nearest-neighbor search (Jegou et al., 2011; Ge et al., 2014; Norouzi & Fleet, 2013). The idea is to decompose the original high-dimensional space into a cartesian product of subspaces that are quantized separately with a joint codebook. To our knowledge, Gong et al. (2014) were the first to introduce these stronger quantizers for neural network quantization, followed by Carreira-Perpiñán & Idelbayev (2017). As we will see in the remainder of this paper, employing this discretization off-the-shelf does not optimize the right objective function, and leads to a catastrophic drift of performance for deep networks.

**Pruning.** Network pruning amounts to removing connections according to an importance criteria (typically the magnitude of the weight associated with this connection) until the desired model size/accuracy tradeoff is reached (LeCun et al., 1990). A natural extension of this work is to prune structural components of the network, for instance by enforcing channel-level (Liu et al., 2017) or filter-level (Luo et al., 2017) sparsity. However, these methods alternate between pruning and re-training steps and thus typically require a long training time.

**Dedicated architectures.** Architectures such as SqueezeNet (Iandola et al., 2016), NASNet (Zoph et al., 2017), ShuffleNet (Zhang et al., 2017; Ma et al., 2018), MobileNets (Sandler et al., 2018) and EfficientNets (Tan & Le, 2019) are designed to be memory efficient. As they typically rely on a combination of depth-wise and point-wise convolutional filters, sometimes along with channel shuffling, they are less prone than ResNets to structured quantization techniques such as PQ. These architectures are either designed by hand or using the framework of architecture search (Howard et al., 2019). For instance, the respective model size and test top-1 accuracy of ImageNet of a MobileNet are 13.4 MB for 71.9%, to be compared with a vanilla ResNet-50 with size 97.5 MB for a top-1 of 76.2%. Moreover, larger models such as ResNets can benefit from large-scale weakly- or semi-supervised learning to reach better performance (Mahajan et al., 2018; Yalniz et al., 2019).

Combining some of the mentioned approaches yields high compression factors as demonstrated by Han *et al.* with Deep Compression (DC) (Han et al., 2016) or more recently by Tung & Mori (Tung & Mori, 2018). Moreover and from a practical point of view, the process of compressing networks depends on the type of hardware on which the networks will run. Recent work directly quantizes to optimize energy-efficiency and latency time on a specific hardware (Wang et al., 2018a). Finally, the memory overhead of storing the full activations is negligible compared to the storage of the weights for two reasons. First, in realistic real-time inference setups, the batch size is almost always equal to one. Second, a forward pass only requires to store the activations of the *current* layer –which are often smaller than the size of the input– and not the whole activations of the network.

# 3 OUR APPROACH

In this section, we describe our strategy for network compression and we show how to extend our approach to quantize a modern ConvNet architecture. The specificity of our approach is that it aims at a small reconstruction error for the outputs of the layer rather than the layer weights themselves. We first describe how we quantize a single fully connected and convolutional layer. Then we describe how we quantize a full pre-trained network and finetune it.

## 3.1 QUANTIZATION OF A FULLY-CONNECTED LAYER

We consider a fully-connected layer with weights $\mathbf{W} \in \mathbf{R}^{C_{\text{in}} \times C_{\text{out}}}$ and, without loss of generality, we omit the bias since it does not impact reconstruction error.

**Product Quantization (PQ).** Applying the PQ algorithm to the columns of $\mathbf{W}$ consists in evenly splitting each column into $m$ contiguous subvectors and learning a codebook on the resulting $mC_{\text{out}}$ subvectors. Then, a column of $\mathbf{W}$ is quantized by mapping each of its subvector to its nearest codeword in the codebook. For simplicity, we assume that $C_{\text{in}}$ is a multiple of $m$, *i.e.*, that all the subvectors have the same dimension $d = C_{\text{in}}/m$.

More formally, the codebook $\mathcal{C} = \{\mathbf{c}_1, \dots, \mathbf{c}_k\}$ contains $k$ codewords of dimension $d$. Any column $\mathbf{w}_j$ of $\mathbf{W}$ is mapped to its quantized version $\mathbf{q}(\mathbf{w}_j) = (\mathbf{c}_{i_1}, \dots, \mathbf{c}_{i_m})$ where $i_1$ denotes the index of the codeword assigned to the first subvector of $\mathbf{w}_j$, and so forth. The codebook is then learned by minimizing the following objective function:

$$\|\mathbf{W} - \widehat{\mathbf{W}}\|_2^2 = \sum_j \|\mathbf{w}_j - \mathbf{q}(\mathbf{w}_j)\|_2^2, \tag{1}$$

where $\widehat{\mathbf{W}}$ denotes the quantized weights. This objective can be efficiently minimized with $k$-means. When $m$ is set to 1, PQ is equivalent to vector quantization (VQ) and when $m$ is equal to $C_{\text{in}}$, it is the scalar $k$-means algorithm. The main benefit of PQ is its expressivity: each column $\mathbf{w}_j$ is mapped to a vector in the product $\mathcal{C} = \mathcal{C} \times \cdots \times \mathcal{C}$, thus PQ generates an implicit codebook of size $k^m$.

**Our algorithm.** PQ quantizes the weight matrix of the fully-connected layer. However, in practice, we are interested in preserving the output of the layer, not its weights. This is illustrated in the case of a non-linear classifier in Figure 1: preserving the weights a layer does not necessarily guarantee preserving its output. In other words, the Frobenius approximation of the weights of a layer is not guaranteed to be the best approximation of the output over some arbitrary domain (in particular for *in-domain* inputs). We thus propose an alternative to PQ that directly minimizes the *reconstruction error* on the output activations obtained by applying the layer to in-domain inputs. More precisely, given a batch of $B$ input activations $\mathbf{x} \in \mathbf{R}^{B \times C_{\text{in}}}$, we are interested in learning a codebook $\mathcal{C}$

that minimizes the difference between the output activations and their reconstructions:

$$\|\mathbf{y} - \widehat{\mathbf{y}}\|_2^2 = \sum_j \|\mathbf{x}(\mathbf{w}_j - \mathbf{q}(\mathbf{w}_j))\|_2^2, \tag{2}$$

where $\mathbf{y} = \mathbf{x}\mathbf{W}$ is the output and $\widehat{\mathbf{y}} = \mathbf{x}\widehat{\mathbf{W}}$ its reconstruction. Our objective is a re-weighting of the objective in Equation (1). We can thus learn our codebook with a weighted $k$-means algorithm. First, we unroll $\mathbf{x}$ of size $B \times C_{\text{in}}$ into $\widetilde{\mathbf{x}}$ of size $(B \times m) \times d$ *i.e.* we split each row of $\mathbf{x}$ into $m$ subvectors of size $d$ and stack these subvectors. Next, we adapt the EM algorithm as follows.

(1) **E-step (cluster assignment).** Recall that every column $\mathbf{w}_j$ is divided into $m$ subvectors of dimension $d$. Each subvector $\mathbf{v}$ is assigned to the codeword $\mathbf{c}_j$ such that

$$\mathbf{c}_j = \underset{\mathbf{c} \in \mathcal{C}}{\operatorname{argmin}} \|\widetilde{\mathbf{x}}(\mathbf{c} - \mathbf{v})\|_2^2. \tag{3}$$

This step is performed by exhaustive exploration. Our implementation relies on broadcasting to be computationally efficient.

(2) **M-step (codeword update).** Let us consider a codeword $\mathbf{c} \in \mathcal{C}$. We denote $(\mathbf{v}_p)_{p \in I_\mathbf{c}}$ the subvectors that are currently assigned to $\mathbf{c}$. Then, we update $\mathbf{c} \leftarrow \mathbf{c}^\star$, where

$$\mathbf{c}^\star = \underset{\mathbf{c} \in \mathbf{R}^d}{\operatorname{argmin}} \sum_{p \in I_\mathbf{c}} \|\widetilde{\mathbf{x}}(\mathbf{c} - \mathbf{v}_p)\|_2^2. \tag{4}$$

This step explicitly computes the solution of the least-squares problem[2]. Our implementation performs the computation of the pseudo-inverse of $\widetilde{\mathbf{x}}$ before alternating between the Expectation and Minimization steps as it does not depend on the learned codebook $\mathcal{C}$.

We initialize the codebook $\mathcal{C}$ by uniformly sampling $k$ vectors among those we wish to quantize. After performing the E-step, some clusters may be empty. To resolve this issue, we iteratively perform the following additional steps for each empty cluster of index $i$. (1) Find codeword $\mathbf{c_0}$ corresponding to the most populated cluster ; (2) define new codewords $\mathbf{c}_0' = \mathbf{c}_0 + \mathbf{e}$ and $\mathbf{c}_i' = \mathbf{c}_0 - \mathbf{e}$, where $\mathbf{e} \sim \mathcal{N}(\mathbf{0}, \varepsilon\mathbf{I})$ and (3) perform again the E-step. We proceed to the M-step after all the empty clusters are resolved. We set $\varepsilon = 1e{-}8$ and we observe that its generally takes less than 1 or 2 E-M iterations to resolve all the empty clusters. Note that the quality of the resulting compression is sensitive to the choice of $\mathbf{x}$.

## 3.2 Convolutional layers

Despite being presented in the case of a fully-connected layer, our approach works on any set of vectors. As a consequence, our apporach can be applied to a convolutional layer if we split the associated 4D weight matrix into a set of vectors. There are many ways to split a 4D matrix in a set of vectors and we are aiming for one that maximizes the correlation between the vectors since vector quantization based methods work the best when the vectors are highly correlated.

Given a convolutional layer, we have $C_{\text{out}}$ filters of size $K \times K \times C_{\text{in}}$, leading to an overall 4D weight matrix $\mathbf{W} \in \mathbf{R}^{C_{\text{out}} \times C_{\text{in}} \times K \times K}$. The dimensions along the output and input coordinate have no particular reason to be correlated. On the other hand, the spatial dimensions related to the filter size are by nature very correlated: nearby patches or pixels likely share information. As depicted in Figure 2, we thus reshape the weight matrix in a way that lead to spatially coherent quantization. More precisely, we quantize $\mathbf{W}$ spatially into subvectors of size $d = K \times K$ using the following procedure. We first reshape $\mathbf{W}$ into a 2D matrix of size $(C_{\text{in}} \times K \times K) \times C_{\text{out}}$. Column $j$ of the reshaped matrix $\mathbf{W}_\text{r}$ corresponds to the $j^{\text{th}}$ filter of $\mathbf{W}$ and is divided into $C_{\text{in}}$ subvectors of size $K \times K$. Similarly, we reshape the input activations $\mathbf{x}$ accordingly to $\mathbf{x}_\text{r}$ so that reshaping back the matrix $\mathbf{x}_\text{r}\mathbf{W}_\text{r}$ yields the same result as $\mathbf{x} * \mathbf{W}$. In other words, we adopt a dual approach to the one using bi-level Toeplitz matrices to represent the weights. Then, we apply our method exposed in Section 3.1 to quantize each column of $\mathbf{W}_\text{r}$ into $m = C_{\text{in}}$ subvectors of size $d = K \times K$ with $k$ codewords, using $\mathbf{x}_\text{r}$ as input activations in (2). As a natural extension, we also quantize with larger subvectors, for example subvectors of size $d = 2 \times K \times K$, see Section 4 for details.

---

[2] Denoting $\widetilde{\mathbf{x}}^+$ the Moore-Penrose pseudoinverse of $\widetilde{\mathbf{x}}$, we obtain $\mathbf{c}^* = \frac{1}{|I_\mathbf{c}|} \widetilde{\mathbf{x}}^+ \widetilde{\mathbf{x}} \left( \sum_{p \in I_\mathbf{c}} \mathbf{v}_p \right)$

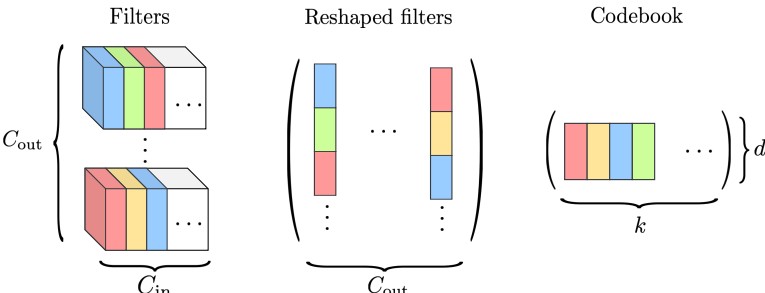

Figure 2: We quantize $C_{\text{out}}$ filters of size $C_{\text{in}} \times K \times K$ using a subvector size of $d = K \times K$. In other words, we spatially quantize the convolutional filters to take advantage of the redundancy of information in the network. Similar colors denote subvectors assigned to the same codewords.

In our implementation, we adapt the reshaping of $\mathbf{W}$ and $\mathbf{x}$ to various types of convolutions. We account for the padding, the stride, the number of groups (for depthwise convolutions and in particular for pointwise convolutions) and the kernel size. We refer the reader to the code for more details.

### 3.3 Network quantization

In this section, we describe our approach for quantizing a neural network. We quantize the network sequentially starting from the lowest layer to the highest layer. We guide the compression of the student network by the non-compressed teacher network, as detailled below.

**Learning the codebook.** We recover the *current* input activations of the layer, *i.e.* the input activations obtained by forwarding a batch of images through the *quantized* lower layers, and we quantize the current layer using those activations. Experimentally, we observed a drift in both the reconstruction and classification errors when using the activations of the non-compressed network rather than the current activations.

**Finetuning the codebook.** We finetune the codewords by distillation (Hinton et al., 2014) using the non-compressed network as the teacher network and the compressed network (up to the current layer) as the student network. Denoting $y_{\text{t}}$ (resp. $y_{\text{s}}$) the output probabilities of the teacher (resp. student) network, the loss we optimize is the Kullback-Leibler divergence $\mathcal{L} = \text{KL}(\mathbf{y}_{\text{s}}, \mathbf{y}_{\text{t}})$. Finetuning on codewords is done by averaging the gradients of each subvector assigned to a given codeword. More formally, after the quantization step, we fix the assignments once for all. Then, denoting $(\mathbf{b}_p)_{p \in I_{\mathbf{c}}}$ the subvectors that are assigned to codeword $\mathbf{c}$, we perform the SGD update with a learning rate $\eta$

$$\mathbf{c} \leftarrow \mathbf{c} - \eta \frac{1}{|I_{\mathbf{c}}|} \sum_{p \in I_{\mathbf{c}}} \frac{\partial \mathcal{L}}{\partial \mathbf{b}_p}. \tag{5}$$

Experimentally, we find the approach to perform better than finetuning on the target of the images as demonstrated in Table 3. Moreover, this approach does not require any labelled data.

### 3.4 Global finetuning

In a final step, we globally finetune the codebooks of all the layers to reduce any residual drifts and we update the running statistics of the BatchNorm layers: We empirically find it beneficial to finetune *all* the centroids after the whole network is quantized. The finetuning procedure is exactly the same as described in Section 3.3, except that we additionally switch the BatchNorms to the training mode, meaning that the learnt coefficients are still fixed but that the batch statistics (running mean and variance) are still being updated with the standard moving average procedure.

We perform the global finetuning using the standard ImageNet training set for 9 epochs with an initial learning rate of $0.01$, a weight decay of $10^{-4}$ and a momentum of $0.9$. The learning rate is decayed by a factor 10 every 3 epochs. As demonstrated in the ablation study in Table 3, finetuning on the true labels performs worse than finetuning by distillation. A possible explanation is that the supervision signal coming from the teacher network is richer than the one-hot vector used as a traditional learning signal in supervised learning (Hinton et al., 2014).

## 4 EXPERIMENTS

### 4.1 EXPERIMENTAL SETUP

We quantize vanilla ResNet-18 and ResNet-50 architectures pretrained on the ImageNet dataset (Deng et al., 2009). Unless explicit mention of the contrary, the pretrained models are taken from the PyTorch model zoo[3]. We run our method on a 16 GB Volta V100 GPU. Quantizing a ResNet-50 with our method (including all finetuning steps) takes about one day on 1 GPU. We detail our experimental setup below. Our code and the compressed models are open-sourced.

**Compression regimes.** We explore a *large block sizes* (resp. *small block sizes*) compression regime by setting the subvector size of regular $3 \times 3$ convolutions to $d = 9$ (resp. $d = 18$) and the sub-vector size of pointwise convolutions to $d = 4$ (resp. $d = 8$). For ResNet-18, the block size of pointwise convolutions is always equal to $4$. The number of codewords or centroids is set to $k \in \{256, 512, 1024, 2048\}$ for each compression regime. Note that we clamp the number of centroids to $\min(k, C_{\text{out}} \times m/4)$ for stability. For instance, the first layer of the first stage of the ResNet-50 has size $64 \times 64 \times 1 \times 1$, thus we always use $k = 128$ centroids with a block size $d = 8$. For a given number of centroids $k$, small blocks lead to a lower compression ratio than large blocks.

**Sampling the input activations.** Before quantizing each layer, we randomly sample a batch of $1024$ training images to obtain the input activations of the current layer and reshape it as described in Section 3.2. Then, before each iteration (E+M step) of our method, we randomly sample $10,000$ rows from those reshaped input activations.

**Hyperparameters.** We quantize each layer while performing $100$ steps of our method (sufficient for convergence in practice). We finetune the centroids of each layer on the standard ImageNet training set during $2,500$ iterations with a batch size of $128$ (resp $64$) for the ResNet-18 (resp. ResNet-50) with a learning rate of $0.01$, a weight decay of $10^{-4}$ and a momentum of $0.9$. For accuracy and memory reasons, the classifier is always quantized with a block size $d = 4$ and $k = 2048$ (resp. $k = 1024$) centroids for the ResNet-18 (resp., ResNet-50). Moreover, the first convolutional layer of size $7 \times 7$ is not quantized, as it represents less than $0.1\%$ (resp., $0.05\%$) of the weights of a ResNet-18 (resp. ResNet-50).

**Metrics.** We focus on the tradeoff between accuracy and memory. The accuracy is the top-1 error on the standard validation set of ImageNet. The memory footprint is calculated as the indexing cost (number of bits per weight) plus the overhead of storing the centroids in float16. As an example, quantizing a layer of size $128 \times 128 \times 3 \times 3$ with $k = 256$ centroids (1 byte per subvector) and a block size of $d = 9$ leads to an indexing cost of $16\,\text{kB}$ for $m = 16,384$ blocks plus the cost of storing the centroids of $4.5\,\text{kB}$.

### 4.2 IMAGE CLASSIFICATION RESULTS

We report below the results of our method applied to various ResNet models. First, we compare our method with the state of the art on the standard ResNet-18 and ResNet-50 architecture. Next, we show the potential of our approach on a competitive ResNet-50. Finally, an ablation study validates the pertinence of our method.

**Vanilla ResNet-18 and ResNet-50.** We evaluate our method on the ImageNet benchmark for ResNet-18 and ResNet-50 architectures and compare our results to the following methods: Trained Ternary Quantization (TTQ) (Zhu et al., 2016), LR-Net (Shayer et al., 2017), ABC-Net (Lin et al., 2017), Binary Weight Network (XNOR-Net or BWN) (Rastegari et al., 2016), Deep Compression (DC) (Han et al., 2016) and Hardware-Aware Automated Quantization (HAQ) (Wang et al., 2018a). We report the accuracies and compression factors in the original papers and/or in the two surveys (Guo, 2018; Cheng et al., 2017) for a given architecture when the result is available. We do not compare our method to DoReFa-Net (Zhou et al., 2016) and WRPN (Mishra et al., 2017) as those approaches also use low-precision activations and hence get lower accuracies, e.g., 51.2% top-1 accuracy for a XNOR-Net with ResNet-18. The results are presented in Figure 4.2. For better readability, some results for our method are also displayed in Table 1. We report the average accuracy and standard deviation over 3 runs. Our method significantly outperforms state of the art papers for

---

[3]https://pytorch.org/docs/stable/torchvision/models

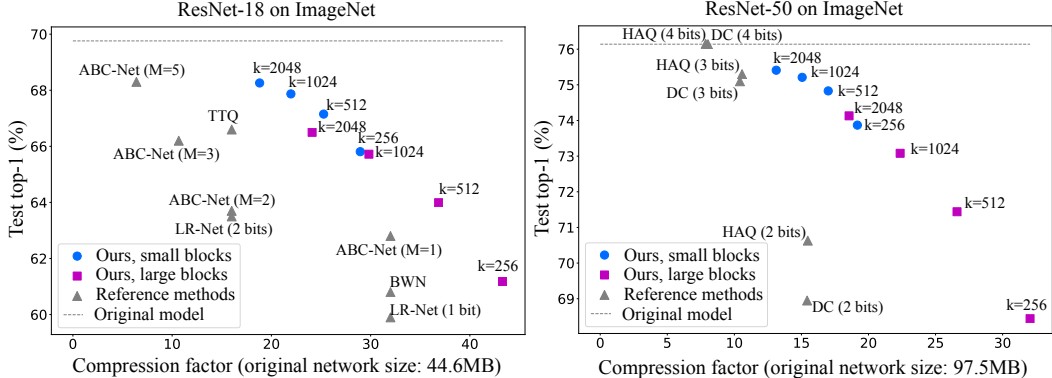

Figure 3: Compression results for ResNet-18 and ResNet-50 architectures. We explore two compression regimes as defined in Section 4.1: small block sizes (block sizes of $d = 4$ and 9) and large block sizes (block sizes $d = 8$ and 18). The results of our method for $k = 256$ centroids are of practical interest as they correspond to a byte-compatible compression scheme.

Table 1: Results for vanilla ResNet-18 and ResNet-50 architectures for $k = 256$ centroids.

| Model (original top-1) | Compression | Size ratio | Model size | Top-1 (%) |
|---|---|---|---|---|
| ResNet-18 (69.76%) | Small blocks | 29x | 1.54 MB | **65.81** ±0.04 |
|  | Large blocks | 43x | 1.03 MB | **61.10** ±0.03 |
| ResNet-50 (76.15%) | Small blocks | 19x | 5.09 MB | **73.79** ±0.05 |
|  | Large blocks | 31x | 3.19 MB | **68.21** ±0.04 |

various operating points. For instance, for a ResNet-18, our method with large blocks and $k = 512$ centroids reaches a larger accuracy than ABC-Net ($M = 2$) with a compression ratio that is 2x larger. Similarly, on the ResNet-50, our compressed model with $k = 256$ centroids in the large blocks setup yields a comparable accuracy to DC (2 bits) with a compression ratio that is 2x larger.

The work by Tung & Mori (Tung & Mori, 2018) is likely the only one that remains competitive with ours with a 6.8 MB network after compression, with a technique that prunes the network and therefore implicitly changes the architecture. The authors report the delta accuracy for which we have no direct comparable top-1 accuracy, but their method is arguably complementary to ours.

**Semi-supervised ResNet-50.** Recent works (Mahajan et al., 2018; Yalniz et al., 2019) have demonstrated the possibility to leverage a large collection of unlabelled images to improve the performance of a given architecture. In particular, Yalniz *et al.* (Yalniz et al., 2019) use the publicly available YFCC-100M dataset (Thomee et al., 2015) to train a ResNet-50 that reaches 79.1% top-1 accuracy on the standard validation set of ImageNet. In the following, we use this particular model and refer to it as semi-supervised ResNet-50. In the low compression regime (block sizes of 4 and 9), with $k = 256$ centroids (practical for implementation), our compressed semi-supervised ResNet-50 reaches **76.12% top-1 accuracy**. In other words, the model compressed to 5 MB attains the performance of a vanilla, non-compressed ResNet50 (vs.97.5MB for the non-compressed ResNet-50).

**Comparison for a given size budget.** To ensure a fair comparison, we compare our method for a given model size budget against the reference methods in Table 2. It should be noted that our method can further benefit from advances in semi-supervised learning to boosts the performance of the non-compressed and hence of the compressed network.

**Ablation study.** We perform an ablation study on the vanilla ResNet-18 to study the respective effects of quantizing using the activations and finetuning by distillation (here, finetuning refers both to the per-layer finetuning and to the global finetuning after the quantization described in Section 3). We refer to our method as Act + Distill. First, we still finetune by distillation but change the quantization: instead of quantizing using our method (see Equation (2)), we quantizing using the standard PQ algorithm and do not take the activations into account, see Equation (1). We refer to this method as No act + Distill. Second, we quantize using our method but perform a standard finetuning using

Table 2: Best test top-1 accuracy on ImageNet for a given size budget (no architecture constraint).

| Size budget | Best previous published method | Ours |
|---|---|---|
| ~1 MB | **70.90%** (HAQ (Wang et al., 2018a), MobileNet v2) | 64.01% (vanilla ResNet-18) |
| ~5 MB | 71.74% (HAQ (Wang et al., 2018a), MobileNet v1) | **76.12%** (semi-sup.ResNet-50) |
| ~10 MB | 75.30% (HAQ (Wang et al., 2018a), ResNet-50) | **77.85%** (semi-sup.ResNet-50) |

Table 3: Ablation study on ResNet-18 (test top-1 accuracy on ImageNet).

| Compression | Centroids $k$ | No act + Distill | Act + Labels | **Act + Distill (ours)** |
|---|---|---|---|---|
| Small blocks | 256 | 64.76 | 65.55 | **65.81** |
| | 512 | 66.31 | 66.82 | **67.15** |
| | 1024 | 67.28 | 67.53 | **67.87** |
| | 2048 | 67.88 | 67.99 | **68.26** |
| Large blocks | 256 | 60.46 | 61.01 | **61.18** |
| | 512 | 63.21 | 63.67 | **63.99** |
| | 1024 | 64.74 | 65.48 | **65.72** |
| | 2048 | 65.94 | 66.21 | **66.50** |

the image labels (Act + Labels). The results are displayed in Table 3. Our approach consistently yields significantly better results. As a side note, quantizing all the layers of a ResNet-18 with the standard PQ algorithm and without any finetuning leads to top-1 accuracies below $25\%$ for all operating points, which illustrates the drift in accuracy occurring when compressing deep networks with standard methods (as opposed to our method).

### 4.3 IMAGE DETECTION RESULTS

To demonstrate the generality of our method, we compress the Mask R-CNN architecture used for image detection in many real-life applications (He et al., 2017). We compress the backbone (ResNet-50 FPN) in the *small blocks* compression regime and refer the reader to the open-sourced compressed model for the block sizes used in the various heads of the network. We use $k = 256$ centroids for every layer. We perform the fine-tuning (layer-wise and global) using distributed training on 8 V100 GPUs. Results are displayed in Table 4. We argue that this provides an interesting point of comparison for future work aiming at compressing such architectures for various applications.

## 5 CONCLUSION

We presented a quantization method based on Product Quantization that gives state of the art results on ResNet architectures and that generalizes to other architectures such as Mask R-CNN. Our compression scheme does not require labeled data and the resulting models are byte-aligned, allowing for efficient inference on CPU. Further research directions include testing our method on a wider variety of architectures. In particular, our method can be readily adapted to simultaneously compress and transfer ResNets trained on ImageNet to other domains. Finally, we plan to take the non-linearity into account to improve our reconstruction error.

Table 4: Compression results for Mask R-CNN (backbone ResNet-50 FPN) for $k = 256$ centroids (compression factor $26\times$).

| Model | Size | Box AP | Mask AP |
|---|---|---|---|
| Non-compressed | 170 MB | 37.9 | 34.6 |
| Compressed | 6.51 MB | 33.9 | 30.8 |

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
