# OpenReview forum: "And the Bit Goes Down: Revisiting the Quantization of Neural Networks"
_ICLR.cc/2020/Conference — Accept (Spotlight)_

### Official Review · AnonReviewer3 · 2019-10-19
**Official Blind Review #3**

**Rating:** 6

**Review:**

This paper addresses to compress the network weights by quantizing their values to some fixed codeword vectors. The authors aim to reduce the distortion of each layer rather than the weight distortion. The proposed algorithm first selects the candidate codeword vectors using k-means clustering and fine-tune them via knowledge distillation. The authors verify the proposed algorithm by comparing it with existing algorithms for ResNet-18 and ResNet-50.

Overall, I think that the proposed algorithm is easy to apply and the draft is relatively well written. Some questions and doubts are listed below.

-In k-means clustering (E-step and M-step), is it correct to multiply \tilde x to (c-v)? I think that the error arising from quantizing v into c is only affected by a subset of rows of \tilde x. For example, if v is the first subvector of w_j, then I think that only 1-st, m+1-th, 2m+1-th, … rows of \tilde x affect to the error.

-Does minimizing reconstruction error minimizes the training loss (before any further fine-tuning) compared to naïve PQ? If not,

-Is there any guideline for choosing the optimal number of centroids and the optimal block size given a target compression rate?

-Is there any reason not comparing the proposed algorithm with other compression schemes? (e.g., network pruning and low-rank approximation)


**Experience Assessment:**

I have read many papers in this area.

**Review Assessment: Checking Correctness Of Derivations And Theory:**

I assessed the sensibility of the derivations and theory.

**Review Assessment: Checking Correctness Of Experiments:**

I assessed the sensibility of the experiments.

**Review Assessment: Thoroughness In Paper Reading:**

I read the paper at least twice and used my best judgement in assessing the paper.

---

> ### Author Response · Authors · 2019-11-08
> **Answer**
>
> We thank Reviewer 3 for raising important questions. We answer them below.
>
> Using \tilde x in the E- and M-steps.
> We agree with Reviewer 3 that “the error arising from quantizing v into c is only affected by a subset of rows of \tilde x”. However, we solve Equation (2) with this proxy algorithm for two reasons. First, using the full \tilde x matrix allows to factor the computation of the pseudo-inverse of \tilde x and thus allows for a much faster algorithm, see answer to Reviewer 2 and the details of the M-step in the paper (as well as footnote 2). Second, early (and slow) experiments suggested that the gains were not significant when using the right subsets of \tilde x in this particular context.
>
> Minimizing the reconstruction error
> Our method results in both better reconstruction error and better training loss than naïve PQ *before* any finetuning. As we state in the paper, applying naive PQ without any finetuning to a ResNet-18 leads to accuracies below 18% for all operating points, whereas our method (without any finetuning) gives accuracy around 50% (not reported in the paper, we will add it in the next version of our paper).
>
> Choosing the optimal number of centroids/blocks size
> There is some rationale for the block size, related to the way the information is structured and redundant in the weight matrices (see in particular point 1 of answer to Reviewer 1). For instance, for convolutional weight filters with a kernel size of 3x3, the natural block size is 9, as we wish to exploit the spatial redundancy in the convolutional filters. For the fully-connected classifier matrices and 1x1 convolutions however, the only constraint on the block size if to be a divisor of the column size. Early experiments when trying to quantize such matrices in the row or column direction gave similar results. Regarding the number of centroids, we expect byte-aligned schemes (256 centroids indexed over 1 byte) to be more friendly for an efficient implementation of the forward in the compressed domain. Otherwise, as can be seen in Figure 3, doubling the number of centroids results in better performance, even if the curve tends to saturate around k=2048 centroids. As a side note, there exists some strategies that automatically adjust for those two parameters (see HAQ for example).
>
> Comparison with pruning and low-rank approximation
> We argue that both pruning and low-rank approximation are orthogonal and complementary approaches to our method, akin to what happens in image compression where the transform stage (e.g., DCT or wavelet) is complementary with quantization. See “Deep neural network compression by in-parallel pruning-quantization”, Tung and Mori for some works investigating this direction.

---

### Official Review · AnonReviewer2 · 2019-10-22
**Official Blind Review #2**

**Rating:** 8

**Review:**

This paper suggests a quantization approach for neural networks, based on the Product Quantization (PQ) algorithm which has been successful in quantization for similarity search. The basic idea is to quantize the weights of a neuron/single layer with a variant of PQ, which is modified to optimize the quantization error of inner products of sample inputs with the weights, rather than the weights themselves. This is cast as a weighted variant of k-means. The inner product is more directly related to the network output (though still does not account for non-linear neuron activations) and thus is expected to yield better downstream performance, and only requires introducing unlabeled input samples into the quantization process. This approach is built into a pipeline that gradually quantizes the entire network.

Overall, I support the paper and recommend acceptance. PQ is known to be successful for quantization in other contexts, and the specialization suggested here for neural networks is natural and well-motivated. The method can be expected to perform well empirically, which the experiments verify, and to have potential impact.

Questions:
1. Can you comment on the quantization time of the suggested method? Repeatedly solving the EM steps can add up to quite an overhead. Does it pose a difficulty? How does it compare to other methods?
2. Can you elaborate on the issue of non-linearity? It is mentioned only briefly in the conclusion. What is the difficulty in incorporating it? Is it in solving equation (4)? And perhaps, how do you expect it to effect the results?

**Experience Assessment:**

I have read many papers in this area.

**Review Assessment: Checking Correctness Of Derivations And Theory:**

N/A

**Review Assessment: Checking Correctness Of Experiments:**

I assessed the sensibility of the experiments.

**Review Assessment: Thoroughness In Paper Reading:**

I read the paper at least twice and used my best judgement in assessing the paper.

---

> ### Author Response · Authors · 2019-11-08
> **Answer**
>
> We thank Reviewer 2 for their support and questions. We answer them below.
>
> Quantization time
> As we state in our paper, quantizing a ResNet-50 (quantization + finetuning steps) takes about one day on one Volta V100 GPU. The time of quantization is around 1 to 2 hours, the rest being dedicated to finetuning. Thus, the time dedicated to quantization is relatively short, especially compared with the fine-tuning and even more with the initial network training. This is because we optimized our EM implementation in at least two ways as detailed below.
> -	The E-step is performed on the GPU (see file src/quantization/distance.py, lines 61-75) with automatic chunking. This means that the code chunks the centroids and the weight matrices into blocks, performs the distance computation on those blocks and aggregates the results. This falls within the map/reduce paradigm. Note that the blocks are automatically calculated to be the largest that fit into the GPU, such that the utilization of the GPU is maximized, so as to minimize the compute time.
> -	The M-step involves calculating a solution of a least squares problem (see footnote 2 in our paper). The bottleneck for this is to calculate the pseudo-inverse of the activations x. However, we fix x when iterating our EM algorithm, therefore we can factor the computation of the pseudo inverse of x before alternating between the E and the M steps (see file src/quantization/solver.py and in particular the docstring).
>
> We provided pointers to the files in the code anonymously shared on OpenReview. To our knowledge, these implementation strategies are novel in this context and were key in the development of our method to be able to iterate rapidly. Both strategies are documented in the code so that they can benefit to the community.
>
> Incorporating the non-linearity
> As the Reviewer rightfully stated, optimally we should take the non-linearity in Equation (4) into account. One could hope for a higher compression ratio. Indeed, the approximation constraint on the positive outputs would stay the same (they have to be close to the original outputs). On the other hand, the only constraint lying on the negative outputs is that they should remain negative (with a possible margin), but not necessarily close to the original negative outputs.  However, our early experiments with this method resulted in a rather unstable EM algorithm. This direction may deserve further investigation.

---

### Official Review · AnonReviewer4 · 2019-10-29
**Official Blind Review #4**

**Rating:** 6

**Review:**

This paper proposes to use codes and codebooks to compress the weights. The authors also try minimizing the layer reconstruction error instead of weight approximation error for better quantization results.
Distillation loss is also used for fine-tuning the quantized weight. Empirical results on resnets show that the proposed method has a good compression ratio while maintaining competitive accuracy.

This paper is overall easy to follow. My main concern comes from the novelty of this paper. The two main contributions of the paper:
(1) using codes and codebooks to compress weights; and
(2) minimizing layer reconstruction error instead of weight approximation error
are both not new. For instance, using codes and codebooks to compress the weights has already been used in [1,2].  A weighted k-means solver is also used in [2], though the "weighted" in [2] comes from second-order information instead of minimizing reconstruction error. In addition, minimizing reconstruction error has already been used in low-rank approximation[3] and network pruning[4].
Clarification of the connections/differences, and comparison with these related methods should be made to show the efficacy of the proposed method.

It is not clear how the compression ratio in table 1 is obtained. Say for block size d=4, an index is required for each block, and the resulting compression ratio is at most 4 (correct me if I understand it wrong).
Can the authors provide an example to explain how to compute the compression ratio?

[1]. Model compression as constrained optimization, with application to neural nets. part ii: quantization.
[2]. Towards the limit of network quantization.
[3]. Efficient and Accurate Approximations of Nonlinear Convolutional Networks.
[4]. ThiNet: A Filter Level Pruning Method for Deep Neural Network Compression.



**Experience Assessment:**

I have published one or two papers in this area.

**Review Assessment: Checking Correctness Of Derivations And Theory:**

I assessed the sensibility of the derivations and theory.

**Review Assessment: Checking Correctness Of Experiments:**

I assessed the sensibility of the experiments.

**Review Assessment: Thoroughness In Paper Reading:**

I read the paper at least twice and used my best judgement in assessing the paper.

---

> ### Author Response · Authors · 2019-11-08
> **Answer**
>
> We thank Reviewer 4 for stating that “the proposed method has a good compression ratio while maintaining competitive accuracy”. We provide clarification for the two main questions of the Reviewer below.
>
> Novelty of the paper
> As we state in our introduction, using codebooks to compress networks is not new, as well as using a weighted k-means technique. However, as we state in the paper: “The closest work we are aware of is the one by Choi et al. (2016), but the authors use a different objective (their weighted term is derived from second-order information) along with a different quantization technique (scalar quantization). Our method targets a better in-domain reconstruction, as depicted by Figure 1”.
>
> Note that we already cite two of the suggested references by Reviewer 4, namely “Towards the limit of network quantization” and “ThiNet: A filter level pruning method for deep neural network compression” in our work. We will further clarify our positioning in an updated version of the paper.
>
> Compression ratio
> We provide an example of the computation of compression ratio in Section 4.1, paragraph “Metrics”. Let us detail it further here. The memory footprint of a compressed layer is split between the indexing cost (one index per block indicating the centroid used to encode the block) and the cost of storing the centroids. Say we quantize a layer of size 128 × 128 × 3 × 3 with 256 centroids and a block size of 9. Then, each block of size 9 is indexed by an integer between 0 and 255: such integer can be stored using 8 bits or 1 byte (as 2^8 = 256). Thus, as we have 128 x 128 blocks, the indexing cost is 128 x 128 x 1 byte = 16,384 bytes = 16 kB. Finally, we have to store 256 centroids of dimension 9 in fp16, which represents 256 x 9 floats (fp16) = 256 x 9 x 2 = 4,608 bits = 4.5 kB. The size of the compressed model is the sum of the sizes of the compressed layers. Finally, we deduce the overall compression ratio which is the size of the compressed model divided by the size of the non-compressed model.

---

### Official Review · AnonReviewer1 · 2019-11-05
**Official Blind Review #1**

**Rating:** 8

**Review:**

The suggested method proposes a technique to compress neural networks bases on PQ quantization. The algorithm quantizes matrices of linear operations, and, by generalization, also works on convolutional networks. Rather than trying to compress weights (i.e. to minimize distance between original and quantized weights), the algorithm considers a distribution of unlabeled inputs and looks for such quantization which would affect output activations as little as possible over that distribution of data. The algorithm works by splitting each column of W_ij into m equal subvectors, learning a codebook for those subvectors, and encoding each of those subvectors as one of the words from the codebook.

The method provides impressive compression ratios (in the order of x20-30) but at the cost of a lower performance. Whether this is a valuable trade-off is highly application dependent.

Overall I find the paper interesting and enjoyable. However, as I am not an expert in the research area, I can not assess how state of the art the suggested method is.

There are a few other questions that I think would be nice to answer. I will try to describe them below:

Suppose we have a matric W_{ij} with dimensions NxM where changing i for a given j defines a column. By definition, linear operation is defined
y_i = sum_j W_ij x_j . Now say each column of matrix W is quantized into m subvectors. We can express W_ij in the following way:
W_ij = (V^1_ij + V^2_ij + ... V^m_ij)x_j where V^m_ij is zero everywhere except for the rows covering a given quantized vector.
For example, if W had dimensions of 8x16 and m=4,
V^2_{3,j}=0, for all j, V^2_{4,j}=non_zero, V^2_{7,j}=non_zero, V^2_{8,j}=0, V^2_{i=4:8,j}=one_of_the_quantized_vectors.

y_i = sum_j W_ij x_j = sum_k sum_j (V^k_ij) x_j =def= sum_k z^k_i where z^k are partial products: z^k_i=0 for i<k*N/m and i>(k+1)N/m

Thus, the suggested solution effectively splits the output vector y_i into m sections, defines sparse matrices V^k_{ij} 1<=k<=m, and performs column-wise vector quantization for these matrices separately.

Generally, it is not ovious or given that the current method would be able to compress general matrices well, as it implicitly assumes that weight W_{ij} has a high "correlation" with weights W_{i+kN/m,j} (which I call "vertical" correlation), W_{i,k+some_number} (which I call "horizontal" correlation) and W_{i+kN/m,k+some_number} (which I call "other" correlation). It is not given that those kind of redundancies would exist in arbitrary weight matrices.

Naturally, the method will work well when weight matrices have a lot of structure and then quantized vectors can be reused. Matrices can have either "horizontal" or "vertical" redundancy (or "other" or neither). It would be very interesting to see which kind of redundancy their method managed to caprture.

In the 'horizontal' case, it should work well when inputs have a lot of redundancy (say x_j' and x_j'' are highly correlated making it possible to reuse code-words horizontally within any given V^k: V^k_ij'=V^k_ij''). However, if thise was the case, it would make more sense to simply remove redundancy by prunning input vector x_j by removing either x_j' or x_j'' from it. This can be dome by removing one of the outputs from the previous layer. This can be a symptom of a redundant input.

Another option is exploiting "vertical" redundancy: this happens when output y_i' is correlated with output y_{i'+N/m}. This allows the same code-word to be reused vertically. This can be a symptom of a redundant output. It could also be the case that compressibility could be further subtantially improved by trying different matrix row permutations. Also, if one notices that y_i' ir correlated with y_i'', it might make sense to permute matrix rows in such a way that both rows would end up a multiple N/m apart. It would be interesting to see how this would affect compressibility.

The third case is when code words are reused in arbitrary cases.

Generally, I think that answering the following questions would be interesting and could guide further research:
1. It would be very interesting to know what kind of code-word reusa patterns the algorithm was able to capture, as this may guide further research.
2. How invariance copressibility is under random permutations of matrix rows (thus also output vectors)?


**Experience Assessment:**

I do not know much about this area.

**Review Assessment: Checking Correctness Of Derivations And Theory:**

I assessed the sensibility of the derivations and theory.

**Review Assessment: Checking Correctness Of Experiments:**

I assessed the sensibility of the experiments.

**Review Assessment: Thoroughness In Paper Reading:**

I read the paper at least twice and used my best judgement in assessing the paper.

---

> ### Author Response · Authors · 2019-11-08
> **Answer**
>
> We thank Reviewer 1 for their insightful questions and suggestions. We agree that Product Quantization (PQ) is key to get “impressive compression ratio” while maintaining competitive accuracy, provided that there is some special structure and redundancy in the weights and the way we quantize them.
>
> Which kind of redundancy does our method capture?
> As rightfully stated by Reviewer 1, choosing which elementary blocks to quantize in the weight matrices is crucial for the success of the method (what the Reviewer calls “horizontal/vertical/other” correlation). In what follows, let us focus on the case of convolutional weights (of size C_out x C_in x K x K). As we state in our paper: “There are many ways to split a 4D matrix in a set of vectors and we are aiming for one that maximizes the correlation between the vectors since vector quantization-based methods work the best when the vectors are highly correlated”. We build on previous work that have documented the *spatial redundancy* in the convolutional filters [1], hence we use blocks of size K x K. Therefore, we rely on the particular nature of convolutional filters to exploit their spatial redundancy. We have tried other ways to split the 4D weights into a set of vectors to in preliminary experiments, but none was on par with the proposed choice.  We agree with Reviewer 1 that the method would probably not yield as good a performance for arbitrary matrices.
>
> Using row permutations to improve the compressibility?
> This is a very good remark. Indeed, redundancy can be artificially created by finding the *right* permutation of rows (when we quantize using column blocks for a 2D matrix). Yet in our preliminary experiments, we observed that PQ performs systematically worse both in terms of reconstruction error and accuracy of the network that when applying a random permutation to a convolutional filter. This confirms that our method captures the spatial redundancy of the convolutional filters as stated in the first point.
>
> [1] Exploiting linear structure within convolutional networks for efficient evaluation, Denton et al.

---

### Public Comment · ~Weihan_Chen1 · 2019-10-25
**similiar work without quoted**

https://arxiv.org/abs/1512.06473

---

> ### Author Response · Authors · 2019-10-25
> **Answer**
>
> Thanks for pointing out this reference! It is definitely relevant to our work, and therefore we will add it in our paper. We would like to point out that our method goes beyond this prior work on several aspects:
> - We use Product Quantization (i.e quantizing chunks of columns) whereas in the cited work the authors use Vector Quantization (i.e. the authors quantize the columns). Our choice takes better advantage of the spatial redundancy of information in the convolutional filters as VQ is less likely to discover the mutual dependency except if using very large amount of data for learning.
> - The cited work does not quantize the layers sequentially and does not finetune the learned centroids -- it finetunes the dense (non-compressed) weights of the classifier.
> - The cited work does not use distillation to take advantage of the teacher, non-compressed network to help compressing the student network.
> - The speed-up reported in the cited work assumes that the scalar products between the input activations of all layers and the centroids are already pre-computed, which is not the case in a real inference scenario.

---

### Public Comment · ~Sourya_Basu1 · 2019-12-20
**Similar theoretical work on quantization of neural networks**

I was wondering if the work in [1] is related to this paper. It seems to me that [1] is similar to this work but seen from a somewhat theoretical point of view using high-rate functional quantization.

[1] Avhishek Chatterjee, and Lav R. Varshney. "Towards optimal quantization of neural networks." 2017 IEEE International Symposium on Information Theory (ISIT). IEEE, 2017.

---

> ### Author Response · Authors · 2019-12-24
> **Answer**
>
> Thanks for pointing out the reference! The authors propose an interesting theoretical viewpoint on quantization, and also consider the activations to derive theoretical bounds on the MSE error when using scalar quantization. We will include this work in our final version.

---

### Author Response · Authors · 2020-02-10
**Final version**

Dear AC, Dear reviewers,

Thank you again for your constructive comments and feedback. We uploaded the final version of our manuscript.

See you in Ethiopia!

---

> ### Public Comment · ~Eunhui_Kim1 · 2020-04-23
> **Question about open code and different accuracy when infer by using quantized pth file**
>
> Dear authors:
>
>  Thank you for your efforts to make improved achivement in the perspective of high compression ratio with high accuracy.
>
>  I tried to validate your code according to your README.md and your paper in my development environment.
>
>  I used the github code https://github.com/facebookresearch/kill-the-bits.
>
>  After quantize with as follow args, I can get pth files per layer and state_dict_compressed.pth, finally.
>
>  Thus using this compressed pth, I ran inference.
>
>  The result accuracy is 10%, however When I used the given compresed pth - 'models/compressed/resnet18_small_block.pth', then it shows the accuracy as your paper inform.
>
>  The args I used for quantization experiment using your code:
>
>  model - resnet18
>  dataset - imagenet
>  n-iter - 100, # of EM iteration
>  n-activations - 1024,  size of the batch of activations
>  block-size-cv - 9, quantization block size for 3x3 conv
>  block-size-pw - 4, quantization block size for 1x1 conv
>  block-size-fc - 4, quantization block size for fully-c layers
>  n-centroids-cv,  256,  # of centroids for 3x3 conv
>  n-centroids-pw,   256,  # of centroids for 1x1 conv
>  n-centroids-fc,  2048, # of centroids for classifier
>  n-centroids-t, 4, threshold for reducing # of centroids
>  eps, 1e-8, empty cluster resolution
>  n-workers, 20, # of workers for data loading
>  finetune-centroids, 2500, # of iters for layer-wise fine tuning of centroids
>  lr-centroids, 0.05, Learning rate to fine tune centroids
>  momentum-centroids, 0.9, momentum when using SGD
>  weight-decay-centroids, 1e-4, weight decay
>  finetune-whole, 10000, # of iters for global fine tuning of centroids
>  lr-whole, 0.01, learning rate to fine tune classifier
>  momentum-whole, 0.9, momentum when using SGD
>  weight-decay-whole, 1e-4, weight decay
>  finetune-whole-epochs, 9, # of epochs for global fine tuning of the centroids
>  finetune-whole-stepsize, 3, learning rate schedule for global fine tuning of the centroids
>  batch-size, 128, batch size for fine-tuning step
>
> The development environment is on the pytorch 1.4.0 version with 32GB V100 2-GPUs.
>
> Could you kindly explain the reason I can not validate your notifed accuracy as your paper?
>
> Thank you

---

> > ### Author Response · Authors · 2020-04-23
> > **Answer**
> >
> > Good morning Eunhui Kim,
> >
> > Thanks for your interest in our work! To facilitate the collaboration and the debugging, could you please fill an issue here: https://github.com/facebookresearch/kill-the-bits/issues/new by copy-pasting what you wrote above? Also, could you indicate in the issue:
> >
> > - Relevant output logs?
> > - Clarify the following: do you refer to the validation accuracy as the one given by running inference.py on the compressed model obtained by running quantize.py? In this case, resolved issue #9 (https://github.com/facebookresearch/kill-the-bits/issues/9)  should help you (you have to patch two lines of code in inference.py).
> >
> > Thanks again for reaching out,
> >
> > The authors

---

### Public Comment · ~Eunhui_Kim1 · 2020-04-24
**It works! I can evaluate the result as your paper.**

Thank you
After I change the code as given in the issue (https://github.com/facebookresearch/kill-the-bits/issues/9), I can get the accuracy as your paper.

---

### Decision · Program_Chairs · 2019-12-19

**Decision:**

Accept (Spotlight)

**Comment:**

This paper addresses to compress the network weights by quantizing their values to some fixed codeword vectors. The paper is well written, and is overall easy to follow. The proposed algorithm is well-motivated, and easy to apply. The method can be expected to perform well empirically, which the experiments verify, and to have potential impact. On the other hand, the novelty is not very high, though this paper uses these existing techniques in a different setting.